



# Harmattan, Saharan heat low and West African Monsoon circulation: Modulations on the Saharan dust outflow towards the north Atlantic

Kerstin Schepanski[1], Bernd Heinold[1], and Ina Tegen[1]

[1]Leibniz Institute for Tropospheric Research (TROPOS), Leipzig, Germany

*Correspondence to:* Kerstin Schepanski (schepanski@tropos.de)

**Abstract.** The outflow of dust from the North African continent towards the north Atlantic is stimulated by the atmospheric circulation over North Africa, which modulates the spatio-temporal distribution of dust source activation and consequently the entrainment of mineral dust into the boundary layer, as well as the transport of dust out of the source regions. The atmospheric circulation over the North African dust source regions, predominantly the Sahara and the Sahel, is characterised by three major

circulation regimes: (1) the Harmattan (trade winds), (2) the Saharan heat low (SHL), and (3) the West African Monsoon circulation. The strength of the individual regimes controls the Saharan dust outflow by affecting the spatio-temporal distribution of dust emission, transport pathways, and deposition fluxes.

This study aims at investigating the atmospheric circulation pattern over North Africa with regard to its role favouring dust emission and dust export towards the tropical North Atlantic. The focus of the study is on summer 2013 (June to August),

during which also the SALTRACE (Saharan Aerosol Long-range TRansport and Aerosol-Cloud interaction Experiment) field campaign took place. It involves satellite observations by the Spinning Enhanced Visible and InfraRed Imager (SEVIRI) flying on-board the geostationary Meteosat Second Generation (MSG) satellite, which are analysed and used to infer a data set of active dust sources. The spatio-temporal distribution of dust source activation frequencies (DSAF) allows for linking the diurnal cycle of dust source activations to dominant meteorological controls on dust emission. In summer, Saharan dust source

activations clearly differ from dust source activations over the Sahel regarding the time-of-day when dust emission begins. The Sahara is dominated by morning dust source activations predominantly driven by the break-down of the nocturnal low-level jet. In contrast, dust source activations in the Sahel are predominantly activated during the second half of the day when down-drafts associated with deep moist convection are the major atmospheric driver. Complementary to the satellite-based analysis on dust source activations and implications from their diurnal cycle, simulations on atmosphere and dust life-cycle were performed

using the meso-scale atmosphere-dust model system COSMO-MUSCAT (COSMO: COnsortium for Small-scale MOdelling; MUSCAT: MUltiScale Chemistry Aerosol Transport Model). Fields from this simulation were analysed regarding the variability of the Harmattan, the Saharan heat low, and the Monsoon circulation as well as their impact on the variability of the Saharan dust outflow towards the north Atlantic. This study illustrates the complexity of the interaction among the three major circulation regimes and their modulation of the North African dust outflow. Enhanced westward dust fluxes frequently appear





following a phase characterised by a deep SHL. Ultimately, findings from this study contribute to the quantification of the interannual variability of the atmospheric dust burden.

# 1 Introduction

Mineral soil particles eroded from bare soils by wind and subsequently mixed into the atmosphere contribute significantly to
the global atmospheric aerosol burden (*Tegen and Schepanski*, 2009). Once suspended in the atmosphere, dust aerosol interacts with solar and terrestrial radiation and is involved in cloud and precipitation forming processes (*Carslaw et al.*, 2010; *Tao et al.*, 2012). Depending on its mineralogical composition, which is determined by the source region, dust transported to remote places may deliver micro-nutrients and thus fertilize terrestrial and marine ecosystems (*Okin et al.*, 2004; *Mahowald et al.*, 2005). Besides its impact on the environment, dust suspended in the air affects the anthroposphere: Dust haze reduces the air
quality and is considered to be responsible for increased cases of respiratory diseases, poor human health, and reduced quality of life in general (*Morman and Plumlee*, 2013).

During the last decades, much effort has been put into research understanding processes involving dust aerosols such as dust radiative feedback, dust cloud interactions, dust (iron) fertilization and the contribution of dust to reduced air quality and associated concerns on human health, technical devices, or infrastructure. For estimating the impact of dust on the Earth system,
knowledge of the atmospheric dust life-cycle consisting of dust source activation and subsequent dust emission, dust transport routes, and dust deposition is crucial. Although dust transport pathways are generally well represented by atmosphere-dust models, the knowledge of the spatial distribution of dust sources and their geomorphological characteristics are still limited although vital for a realistic simulation of the full atmospheric dust life-cycle (*Schepanski et al.*, 2013). Today, a diversity of atmosphere-dust model systems are available for the LES-scale (e.g., *Klose and Shao*, 2013), the regional scale (e.g., *Heinold*
*et al.*, 2011; *Chaboureau et al.*, 2016), and the global scale (e.g., *Ridley et al.*, 2012), which all together are illustrating the range of uncertainty in dust concentration distributions. For North Africa, the dust model intercomparison study carried out in the framework of AeroCom (*Huneeus et al.*, 2011) suggests a range of annual dust emission fluxes of 400 to 2,200 Tg year$^{-1}$. Recent studies analysing satellite observations provide new insights in the spatio-temporal distribution of active dust sources and thus allow for a detailed analysis of their variability in space and time (*Schepanski et al.*, 2009b; *Tegen et al.*, 2013; *Wagner*
*et al.*, 2016). Thereby, the temporal resolution of the satellite data is crucial as a temporal off-set between dust source activation and satellite overpass may lead to a misplacement of dust sources (*Schepanski et al.*, 2012; *Kocha et al.*, 2013). An as accurate as possible inferred spatial distribution of dust sources allows for examining geomorphological units (*Crouvi et al.*, 2012) and mineralogical characteristics (*Formenti et al.*, 2014) of dust sources. These characteristics ultimately determine the nature of the dust source in terms of sediment supply and susceptibility to interannual variability, the contribution to the dust burden in
remote locations (*Banks et al.*, 2017), and the optical and micro-physical properties of eroded dust aerosols (*Carslaw et al.*, 2010). A map of dust source activations including the time indicating the onset of dust emission as published by *Schepanski et al.* (2007) further allows to investigate the meteorological controls on dust emission and to quantify the role of frequently recurring synoptic-scale atmospheric features for frequently active dust sources. From such studies, the nocturnal low-level



jet (LLJ) was found to be the dominant driver for dust source activation over North Africa (*Schepanski et al.*, 2009b), which was verified by several follow-up studies (e.g., *Fiedler et al.*, 2013; *Tegen et al.*, 2013; *Fiedler et al.*, 2014). Besides the LLJ, deep moist convection embedded in the intertropical discontinuity region fosters the formation of dust fronts by cold pools in particular over the Sahel zone during the wet season (*Bou Karam et al.*, 2009; *Heinold et al.*, 2013; *Fiedler et al.*, 2014). The

advection of cooler air from the Mediterranean Sea into the Sahara, occasionally reaching the Sahel zone, may promote the formation of a Soudano-Saharan Depression. This depression then migrates westwards, eventually turns anti-cyclonically into a northward direction and may be enhanced by Atlas lee-cyclogenesis ultimately forming a Mediterranean cyclone (*Schepanski et al.*, 2011).

Various measurement campaigns and modelling efforts have been carried out investigating the atmospheric dust cycle and

related processes in order to improve the general understanding of dust suspended in the atmosphere and related atmospheric processes. With focus on Saharan dust and related dust radiation feedbacks and cloud formation processes, several projects focussed on the North African continent and its adjacent dust outflow regions during the last decade: The SAMUM-1 experiment (Saharan Mineral Dust Experiment, *Heintzenberg* (2009)), which was conducted in Southern Morocco during May and June 2006 and aimed for characterizing Saharan dust particles to quantify dust-related radiative effects; the SAMUM-2 experiment

(*Ansmann*, 2011), which took place during January and February 2008 at Cape Verde in the so-called dust outflow region, and focused on the optical properties of aged dust and dust aerosol mixed with biomass-burning aerosol; the DODO (Dust Outflow and Deposition to the Ocean, *McConnell et al.* (2008)) experiments aiming at determining dust radiative properties over the dust outflow region in 2006; the Fennec project (*Ryder et al.*, 2015) aiming at improving the understanding of physical processes that control the Saharan climate system and conducting field experiments during June and July 2011 and 2012; and

the SALTRACE experiment, which was conducted in June to July 2013 and investigated the dust properties related to dust radiative effects and micro-physical processes of aged Saharan dust that has been transported to the Caribbean (*Weinzierl*, in press). Although many studies focus on the continental region including the adjacent dust outflow, measurements during SALTRACE were obtained at both sides of the tropical Atlantic: at Cape Verde relatively close to the North African continent and at Barbados situated in the Caribbean Sea.

This study focuses on North African dust source activations and their controlling on atmospheric circulation patterns that ultimately determine dust transport towards the Caribbean Sea. It was carried out in the framework of the SALTRACE project and contributes to the understanding on the origin of dust aerosol transported to and eventually measured in the Caribbean (*Weinzierl*, in press). The aim of this study is twofold. First, it discusses the spatio-temporal distribution of active dust sources over North Africa regarding implicit information on the meteorological elements driving dust emission and their predomi-

nance during a typical summer season. To achieve this, the study makes use of satellite observations for identification of active dust source regions. Second, the study elaborates the atmospheric circulation with regard to its relevance for dust source activation and dust transport towards the tropical North Atlantic. For this, simulations from the dust-atmosphere model system COSMO-MUSCAT are analysed. The period of investigation is chosen to cover the entire summer season June to August 2013. Transport-related processes occurring during dust transport over the North Atlantic are presented in a companion paper.

The manuscript is structured as follows: Following a general introduction to atmospheric controls on dust source activation in



Section 2, an overview on the data sets used for the identification of dust sources and characterization of connected atmospheric circulation patterns is given in Section 3. Section 4 presents the spatio-temporal variability in dust source activation and leads to the analysis of different atmospheric features modulating the North African dust burden in Section 6. Aspects addressed therein are discussed in Section 7 with regard to its impact on dust export fluxes, and summarized in Section 8.

## 2 Atmospheric controls on dust source activation

### 2.1 Dust source characteristics

Dust emission is a threshold problem: Momentum provided by the wind is an essential prerequisite to mobilize soil particle from bare ground. Thereby, the minimum amount of momentum required for particle mobilization depends on surface characteristics such as soil texture and particle size distribution (*Kok et al.*, 2012). However, dust mobilization may also be limited by the soil erodibility describing the susceptibility of the surface for wind erosion. The soil erodibility is determined by (1) roughness elements such as vegetation, rocks or soil clods, (2) intrinsic characteristics such as soil texture, mineralogy, or soil content of organic matter, and (3) temporally varying characteristics such as soil moisture, soil aggregation, crusting and the availability of erodible material (*Webb and Strong*, 2011). All characteristics in concert determine the interparticle cohesive forces and thus the amount of momentum required for particle mobilization. Consequently, wind speed distribution and surface sediment characteristics together determine the dust emission flux. On a daily basis, in particular the supply of momentum and thus the wind speed distribution is the dominant factor controlling dust source activation and subsequent dust emission, ultimately resulting into its diurnal cycle (*Schepanski et al.*, 2009b). Additionally, precipitation and subsequent increase in soil moisture control dust emission in particular in the Sahel zone and the northern margins of the Sahara where convective precipitation associated with meso-scale convective systems (MCS) respectively cyclogenesis occurs (*Fiedler et al.*, 2014; *Bergametti et al.*, 2016). Vegetation cover and its changes contribute significantly to the dust variability, in particular at seasonal time scales in regions with distinct vegetation periods.

The focus of this study is on the atmospheric controls on dust source activation and subsequent dust emission. It discusses different synoptic-scale features of the atmospheric circulation affecting the local wind speed distribution and thus inhibiting or forcing dust source activation, which ultimately result in a local variability in the occurrence frequency of dust source activation and consequently atmospheric dust concentrations downwind.

### 2.2 Atmospheric circulation over North African dust source areas

Besides the availability of sediments susceptible to wind erosion, the presence of sufficiently high wind speeds is the most limiting factor for dust emission. Although local wind speeds determine local dust mobilization and subsequent entrainment into the atmospheric boundary layer, the atmospheric circulation provides the general conditions for the state of the atmosphere at local scale. For example, atmospheric turbulence, static stability, wind speed distribution and gustiness are affected by pressure



gradients determining the prevailing geostrophic wind, and advection of humidity and temperature.

The atmospheric circulation over North Africa in boreal summer (June, July, August (JJA)) is dominated by three major features: (1) the Harmattan wind, (2) the Saharan Heat Low (SHL), (3) the West African Monsoon (WAM) circulation including the African Easterly Waves (AEWs), which are characterised by distinct troughs and ridges. The Harmattan names the

north-easterly trade winds over North Africa that result from the continental scale pressure gradient between the subtropical subsidence zone and the inter-tropical convergence zone (ITCZ). With regard to dust entrainment, these winds are of particular interest as they are involved in the development of the nocturnal LLJs, which are the dominant and frequent drivers for dust source activation over North Africa (*Schepanski et al.*, 2009a, b; *Fiedler et al.*, 2013; *Schepanski et al.*, 2015a). As part of the WAM, cool and moist air is transported northward from the Gulf of Guinea into the North African continent. The resulting

baroclinicity fosters the generation of nocturnal LLJ embedded in the monsoonal flow at the southern margins of the Saharan heat low (*Parker et al.*, 2005a; *Bou Karam et al.*, 2008; *Schepanski et al.*, 2009a; *Fiedler et al.*, 2013). The atmospheric environment of moist monsoonal air further allows deep moist convection and the development of Mesoscale Convective Systems (MCSs), in particular along the AEJ that forms a wave-structure over the Sahel zone in summer. The SHL plays an important role for the strength of the monsoon trough (*Thorncroft and Blackburn*, 1999). Dust emission related to deep-moist convection

also occurs along the Saharan side of the Atlas mountains, where orographically induced cold pools or lee-cyclogenesis result into the generation of dust fronts (*Knippertz et al.*, 2009; *Reinfried et al.*, 2009; *Schepanski et al.*, 2009b, 2011; *Bou Karam et al.*, 2010).

**The Saharan Heat Low as connecting element in between the Harmattan and Monsoon circulation**

The Saharan heat low (SHL) is a thermal low formed by intense solar heating of the desert land surface. The heated land surface re-emits thermal energy into the lower atmosphere which subsequent leads to heated air layers. The heated air ascends due to its buoyancy and a zone of low pressure forms at near surface levels. As the SHL is a heat low, there are no fronts associated. As proposed by *Lavaysse et al.* (2009), the low-level atmospheric thickness (LLAT) can be applied as a measure to identify the geographical position, the extent, and strength of the SHL. Thereby, the LLAT is related to the mean temperature

of the atmospheric layer confined by the 700 hPa and 925 hPa and thus represents the heat-induced dilatation of the lower atmospheric levels.

The SHL is suggested to act as a key element of the atmospheric circulation over North Africa. During summer, it is situated over West Africa between the Atlas Mountains to the north and the Hoggar Massif to the east (*Lavaysse et al.*, 2009). Due to its geographical position, its cyclonic circulation affects the strength of the Harmattan winds and the monsoon flow: Along

its eastern flank the south-westerly monsoon flow is increased, along its western flank the north-easterly Harmattan winds are increased (*Lavaysse et al.*, 2009; *Parker et al.*, 2005b). The SHL may even affect the position of the AEJ, and as the temperature gradient between the hot and dry Sahara and the cold and moist Gulf of Guinea impacts on the state of the AEJ. The SHL may even interact with the development of AEWs and the position of the monsoon front (*Thorncroft and Blackburn*, 1999; *Chauvin*, 2010; *Lavaysse et al.*, 2010). The SHL also determines the ventilation of the North African continent through cooler,




maritime air masses that are advected to the Saharan desert due to increased pressure gradients. On its eastern flank, cooler air masses originating from the Mediterranean Sea are transported into the North African continent eventually contributing to the atmospheric moisture budget over the Sahel (*Vizy and Cook*, 2009). On its western flank, air masses originating from the Atlantic ventilate the continent driven by the pressure gradient between the subtropical high (Azores high) and the SHL

(*Grams et al.*, 2010). Furthermore, *Roehrig et al.* (2011) found a modulation of the structure of the Azores high by the so-called pulsation of the SHL. The pulsation of the SHL characterizes the migration of the centre of the SHL from a so-called eastern phase to a so-called western phase (*Chauvin*, 2010; *Roehrig et al.*, 2011). The western phase is associated with higher surface temperatures over the coastal region of Morocco and Mauritania, and a general south-westward propagation of this temperature signature. Lower surface temperatures are evident over the region between Sicily and Libya; this signature is

propagating in south-easterly direction. The eastern phase is associated with the opposite temperature signature. As outlined above, the position (eastern phase versus western phase, *Chauvin* (2010)), the depth and extent of the SHL generally interacts and affects the large-scale circulation over North Africa. In particular, the Harmattan and monsoon flow modulates the nature of the SHL and vice versa (*Lavaysse et al.*, 2009).

Although dust source activation and subsequent dust emission is driven by local wind speed conditions, large scale circulation

determines the local atmospheric conditions fostering or inhibiting dust emission. Modulated by and interacting with Harmattan and Monsoon circulation, the SHL is an atmospheric element that on the one side can be characterised by its depth and extent, and on the other side which reflects implicitly both, the Harmattan and the Monsoon. In this study, we use the method proposed by *Lavaysse et al.* (2009) to identify the position, extent and depth of the SHL from COSMO-MUSCAT geopotential and temperature fields at 700 hPa and 925 hPa level. The area of the SHL is refined to the 90th percentile of the LLAT values.

**2.3   Dust export towards the North Atlantic**

Dust transport towards the tropical Atlantic and across to the Caribbean Sea is determined by the dust conditions over West Africa, the atmospheric circulation advecting the dusty air mass to the coastal regions off West Africa, the entering of the Atlantic region and the formation of the so-called Saharan Air Layer across the Atlantic ocean (e.g., *Schepanski et al.*, 2009a). Although dust emission is the vital element of the dust's journey across the Atlantic, the release of dust-loaded air from the

North African continent to the tropical Atlantic is not inevitable. Dust may also be deposited over the North African continent. Airborne dust circulating over Western Africa may become sucked into the SHL as described by *Ashpole and Washington* (2013) or form the ridge of an AEW (*Jones et al.*, 2003). In both cases, dust may eventually be pushed towards the Atlantic and released as a dust plume once the SHL is in its western phase (*Chauvin*, 2010) or an AEW enters the Atlantic (*Jones et al.*, 2003). Besides that, dust-loaded air masses are transported towards the Atlantic by the Harmattan flow. In that case, the

dust export flux is rather continuous compared to the plume-like structure of dust export associated with the intermittent nature of the SHL pulsation or the passage of AEWs. In summary, SHL and AEW can be understood as a modulation on the North African dust export within the Saharan air layer.



## 3  Data sets providing information on the atmospheric dust life-cycle

To elaborate various concepts of controlling mechanism on the atmospheric dust life-cycle, in particular dust emission and transport over the North African continent, this study combines two different data sets: (1) the DSAF (Dust Source Activation Frequency) data set providing information on the spatio-temporal distribution of active dust sources inferred from satellite

observations, and (2) fields from numerical simulations of the atmospheric dust-life cycle using the atmosphere-dust model system COMSO-MUSCAT.

### 3.1  Dust source identification

Satellite observations as these obtained from the geostationary MSG (Meteosat Second Generation) SEVIRI (Spinning-Enhanced Visible and Infra-Red Imager) instrument at 15-minute temporal resolution provide convenient data for identifying active dust

source. As described by *Schepanski et al.* (2007), the infrared dust index, which is a red-green-blue (RGB) composite image of different combinations of the brightness temperature of the three SEVIRI channels centred at $8.7\mu m$, $10.8\mu m$ and $12.0\mu m$ wavelengths, is used to identify active dust source in terms of dust entrainment into the atmospheric boundary layer. Active dust sources are recorded on a $1° \times 1°$ grid covering North Africa north of $10°$N building up the DSAF data set.

The dust index provides qualitative information on the presence of airborne dust and has previously been used to locate active

dust source within the viewing field of SEVIRI over North Africa (*Schepanski et al.*, 2007, 2009b, 2012) and southern Africa (*Vickery et al.*, 2013). As based on measurements at infrared wavelength, the index is sensitive to atmospheric humidity, which affects the identification of dust layer in regions with high atmospheric humidity such as cold pool outflows (*Brindley et al.*, 2012). However, due to the visual identification of dust plumes the impact of this caveat is minimized compared to automatic retrieval algorithm. The sensitivity against colour shades and propagating plumes is much higher and provides reliable iden-

tification. As for most satellite-based observations, no information on the presence of dust is available underneath clouds. Consequently, dust uplift in the vicinity of deep moist convection may remain unrecorded and may result into an underestimation of dust source activations driven by deep moist convection.

Besides the geographical location of active dust sources, the time of day when the dust source became active was recorded and binned at hourly resolution. In case of several dust source activation per $1° \times 1°$ grid cell, the earliest one determined the

recorded time. A comparison of dust source data sets based on 15-minute resolution satellite observation and daily observations has pointed towards a shift in dust source regions identified in downwind direction due to the temporal off-set between onset of dust source activation and satellite overpass (*Schepanski et al.*, 2012).

### 3.2  COSMO-MUSCAT dust simulations

Fields describing the state of the atmosphere and the dust aerosol distribution are taken from the meso-scale atmosphere-

dust model system COSMO-MUSCAT (COSMO: COnsortium for Small-scale MOdelling; MUSCAT: MUltiScale Chemistry Aerosol Transport Model), which is a non-hydrostatic atmosphere model (COSMO version 5.0) that is coupled on-line to the 3D chemistry tracer transport model MUSCAT. Schemes on dust emission (*Tegen et al.*, 2002) and deposition (*Berge*, 1997;





*Jakobson et al.*, 1997; *Zhang et al.*, 2001) are implemented in MUSCAT and on-line driven by atmospheric and hydrological fields calculated by COSMO (*Heinold et al.*, 2011). Thus, the full atmospheric dust cycle consisting of dust mobilization and emission, transport, and dry/wet deposition is represented by the COSMO-MUSCAT simulations. Five size bins ranging from 0.1 to $48 \mu m$ are defined to resolve the size dependency of dust transport and deposition. Depending of the state of the

atmosphere, in particular the near-surface wind distribution, dust emission fluxes are calculated for regions identified as active dust source using the dust source activation frequency (DSAF) mask inferred from MSG SEVIRI observations as described by *Schepanski et al.* (2007). By applying this DSAF mask, all grid cells, where within the time period 2006-2012 dust source activation was observed at least twice, were treated as potential dust sources. The dust flux then is calculated using the actual wind fields and atmospheric/hydrological conditions calculated by COSMO and considering soil texture and soil size distribution

as proposed by *Tegen et al.* (2002). Additionally, the aerodynamic roughness length $z_0$ is implemented for adjusting the dust emission scheme to the near-surface wind speed distribution provided by COSMO. For grid cells where the maximum monthly average of the leaf area index (LAI) is less than 0.1 and the standard deviation of the sub-grid scale orography is larger than 50m the roughness length from *Prigent et al.* (2012) is taken, elsewhere $z_0$ is set to the optimum value of $z_0 = 0.001$cm to more accurately represent dust emission from sources located within the mountain foothills and the Sahel. This model setup has

already been successfully applied and validated against several observation data sets as described in *Schepanski et al.* (2009a); *Heinold et al.* (2011); *Tegen et al.* (2013); *Schepanski et al.* (2015b, 2016).

COSMO-MUSCAT simulations are performed for the period 15 May to 31 August 2013. The simulation period is chosen to start mid-May to allow for a realistic atmospheric dust distribution for the period of interest (June - August 2013). The model domain is chosen to capture dust source regions contribution to the dust export from North Africa towards the Caribbean and

its transport pathways. Thus, COSMO-MUSCAT simulations at 28 km horizontal grid spacing and 40 vertical $\sigma$-p levels are performed for the domain spanning between 10.0° S, 70.0° W and 40.5° N, 38.8° E . The domain covers west Africa including the parts of the Sahara relevant for westward dust export, and the tropical north Atlantic as far as the Caribbean. The lowest level is centred at 10m above ground.

## 4 North African dust source activity during June and July 2013

Active dust sources were inferred from MSG SEVIRI IR dust index images for the time period 1 June to 31 July 2013 as outlined in section 3. The two-month period covers the SALTRACE field campaign, which took place from 10 June to 14 July 2013. Whereas the dust measurements obtained over Barbados characterise the last part of the atmospheric dust cycle and thus the final stage of the life-cycle of dust in the atmosphere, an encompassing analysis includes the discussion of dust sources

and dust emission processes. E.g., *Groß et al.* (2015) identified different phases of atmospheric dustiness over the Caribbean and Barbados, which were characterized by differing optical properties and atmospheric aerosol layer structures. These phases were related to active dust source regions in order to investigate the relation between dust source characteristics, dust transport route and arrival over the Caribbean.





The combination of time and place of dust source activation thus states that all conditions for dust emission were fulfilled. Here we focus on locally occurring wind speeds that are sufficiently high to foster dust entrainment. The diurnal wind speed distribution is determined by the large-scale atmospheric circulation modulated by smaller-scale processes such as regional wind regimes and local boundary layer processes. Prominent examples for large-scale features are the Harmattan or WAM cir-

culation; regional wind regimes are for example the mountain and valley breeze, but also the formation of LLJs. As topography and atmospheric circulation, which determine the local wind speed distribution and thus the frequency of fulfilling the major atmospheric precondition for dust emission, are not equally distributed over North Africa, spatial and temporal variabilities in dust source activation are expected.

The analysis of the spatial distribution of frequently active dust sources as shown in Fig. 1a illustrates that frequently active

dust sources that also dominate the June-July 2013 period are located within the mountain regions over North Africa besides the dust sources located in the Bodélé Depression. In particular, dust source embedded in the mountain foothills of the Hoggar-Adrar-Air Massif become frequently active during this period. Dust sources with pronounced activity are also observed over the Al Jabel Akhadra Massif in Libya and the southern fringe of the Atlas Mountains. Also, dust sources over southern Mauritania are found to be active. The predominance of dust source embedded in mountain regions agrees with earlier findings based on

the DSAF data base (i.e., *Schepanski et al.*, 2009b, 2012).

Comparing the June-July 2013 period to the four-year June-July 2006-2009 period (Fig. 1b), the patterns of the spatial distribution of DSAF are matching. However, from a broader perspective, the level of occurrence frequency is generally higher for the 2006-2009 period than for 2013, but the maximum is higher for 2013. This can be explained by the overall high DSAF during 2007-2008 as discussed by *Tegen et al.* (2013) and *Wagner et al.* (2016). Also, the Bodélé Depression is significantly

more active during 2013 than compared to the 2006-2009 mean.

The temporal distribution of dust source activation, in particular its diurnal cycle, provides implicitly information on the meteorological driver fostering dust source activation and subsequent dust entrainment into the atmosphere (*Schepanski et al.*, 2009b). Dust source activations during the first half of the day are predominantly linked to the break-down of the nocturnal LLJ, whereas dust source activations occurring during the second half of the day are predominantly related to (moist) convec-

tion (*Schepanski et al.*, 2009b). The climatology of the occurrence frequency of the LLJ shows in particular high frequencies where the Harmattan is the dominating wind regime, and over region of maritime air inflow such as the Atlantic and Mediterranean ventilation areas, and along the northern front of the WAM (cf. *Schepanski et al.*, 2009b; *Fiedler et al.*, 2013). Moist convection and related MCSs develop more frequently towards the south, where the arid Sahara transition into the semi-arid Sahel zone. With northward propagation of the monsoon front, MCSs frequently migrate into the desert, where downdrafts

generate dust storms, which are frequently associated with the formation of impressive dust fronts - the so-called Haboobs.

Figure 2 present the dust source activation frequency for North Africa and thus enables to identify major dust source region. The figure shows the DSAF separately for the two halves of the day: Figs. 2a and b present only these beginning there emission during the first half of the day. Thereby, Fig. 2a depicts the fraction part of all dust source activations, whereas Fig. 2b illustrates the occurrence frequency. Both plots together show first the absolute relevance of individual dust source regions, and second

the relative importance depending on the time. DSAF for the second half of the day are shown similarly on Figs. 2c and d.



Generally, dust emission over North Africa predominantly sets on during the first half of the day, except for some dust source regions in the Sahel zone. A north-south gradient with regard to the time of the day of dust source activation is evident. Dust source regions being activated during the 00-12 UTC time slot dominate both, in relative and absolute frequencies. The second half of the day (12-00 UTC) is dominated by dust source activations in particular over the Sahel and the Mediterranean coastal

region (northern tip of Libya) to the north of the Sahara. One prominent area representing this influence of the northward propagation of the monsoon front is the region south-west of the Hoggar-Adra-Air mountain region. That region is characterised by enhanced absolute occurrence frequencies during the 12-00 UTC time slot.

## 5   Atmospheric dust distribution: Validation of dust aerosol optical depth

The atmospheric dust load is often used to assess the different elements of the atmospheric dust life-cycle consisting of (1)

dust emission, (2) dust transport, and (3) dust deposition. *Dust emission* is determined by source characteristics (i.e. sediment supply and availability) and wind strength, and thus dust emission fluxes are a representative for the connection of these factors. *Dust transport* results from prevailing wind regimes, but also particle buoyancy, which is influenced by atmospheric stability respective turbulence. By validating dust loading provided by numerical simulations against in-situ measurements, the model's ability on balancing out dust emission flux, transport capacity of the atmosphere, and deposition flux is verified. *Dust deposition*

summarizes all dust removal processes. The model's ability to correctly represent local deposition fluxes implies the correct representation of dust emission and transport. As local in-situ measurements of dust emission and deposition fluxes require an extensive measurement infrastructure, measurements of the atmospheric dust load appear a realistic compromise. As the atmospheric dust load is modulated by both the dust emission and deposition fluxes, this parameter is assumed as representative for the entire atmospheric dust life-cycle.

Similarly, dust aerosol optical depth (AOD) representing the total atmospheric column dust load calculated from COSMO-MUSCAT simulations (cf. *Schepanski et al.* (2016) for a description on the calculation) are compared against AOD estimates calculated from sun-photometer measurements, which are organized and made available through the AERONET (AErosol RObotic NETwork, *Holben et al.* (1998)) initiative. AODs for five different stations directly influenced by Saharan dust plumes are selected: Santa Cruz located on the isle of Tenerife (28.473°N 16.247°W, Spain) off the coast of northwest Africa, Capo

Verde on the isle of Sal (16.733°N 22.935°W, Cape Verde) off the coast of West Africa, Dakar (14.394°N 16.959°W, Senegal) in the western-most part of West Africa, and Cinzana (13.278°N 5.934°W, Mauritania) and Banizoumbou (13.541°N 2.665°E, Niger) situated in the Sahel and bordering to the southern margin of the Sahara. The five stations are located downwind the source regions and within the dust transport path towards the Atlantic. A more thorough validation of the model simulation including satellite products is published in *Schepanski et al.* (2016), which discusses Saharan dust transport towards the

Mediterranean basin.

As the COSMO-MUSCAT AODs consider dust aerosols only in these simulations analysed here, sun-photometer measurements in contrast are affected by all types of aerosol particles (i.e., mineral dust, soot, sea spray aerosol), coarse-mode AODs (*O'Neill et al.*, 2003) predominantly represent the dust fraction. This is a common approach in order to allow for the best





comparability with model simulations where mineral dust is the only simulated type of aerosol (*Tegen et al.*, 2013; *Schepanski et al.*, 2015b, 2016). The comparison of COSMO-MUSCAT dust AODs with AERONET coarse mode AODs as shown in Fig. 3 depicts the model's ability to simulate the temporal variability of the atmospheric dust load over the particular sites. Generally, the model represents the range of AOD values and the temporal evolution of the coarse mode AOD derived from

sun-photometer measurements at these sites well, although some minor difference on single event basis are evident. This suggests that COSMO-MUSCAT sufficiently well captures the atmospheric dust life-cycle including its relevant and determining atmospheric processes. In particular, matching dust AODs suggest that COSMO-MUSCAT is able to capture the meteorology correctly and, furthermore, to balance dust emission dust removal fluxes sufficiently good. The model results therefore are trustworthy for deriving the relations between the dust cycle and the drivers of atmospheric circulation in the study period.

## 6 Atmospheric circulation modulating the atmospheric dust burden over North Africa

In the following subsections, the role of the Harmattan, the SHL and the WAM will be examined exemplarily for summer (June-August) 2013 with regard to their impact on the atmospheric dust load and dust transport towards the north Atlantic.

### 6.1 Harmattan

The strength of the Harmattan, which names the trade winds over North Africa, is determined by the pressure gradient between

the subtropics and the tropics. Due to the geographic situation, the gradient is strongest over the North African continent. The north-easterly Harmattan winds are deflected by local topography and modulated by variability in pressure distribution. *Rodriguez et al.* (2015) introduced the North African dipole intensity (NAFDI) index in order to represent the temporal variability in the Harmattan flow. The index is similar to the North Atlantic Oscillation (NAO) index, but represents the pressure gradient over the North African continent, which is of direct relevance for the strength of the Harmattan winds. The NAFDI is the

normalized difference of the anomaly of the geopotential $\Phi$ at 700hPa between two $3° \times 3°$ boxes located north and south of the Sahara desert. For this study, the NAFDI index $F$ is calculated as follows:

$$F = \frac{1}{10} \left( \left( \Phi_n^t - \bar{\Phi}_n \right) - \left( \Phi_s^t - \bar{\Phi}_s \right) \right), \tag{1}$$

with $\Phi_n$ denoting geopotential averaged over a box located north of the Sahara (30-32°N, 5-7°W), and $\Phi_s$ representing the geopotential averaged over a box located south of the Sahara (10-13°N, 6-8°W). The index $t$ denotes the date out of the period

1 June to 31 August 2013, as the NAFDI values are calculated on a daily basis for this study. Temporal means $\bar{\bar{\Phi}}$ are calculated over the entire period, here JJA 2013.

The temporal evolution of the NAFDI index as shown in Fig. 4a for June to August 2013 reflects the intensity of the pressure gradient across the Sahara and thus the strength of the Harmattan winds. The JJA 2013 summer season is characterised by low NAFDI index values during the first three weeks. On 24 June the index switches from negative values to positive values

and remains positive until 24 August only with brief intermittence. Regarding the strength and thus the predominance of the



Harmattan flow, JJA 2013 can roughly be separated into three phases: Phase I representing the period from 1 to 23 June, which is characterised by negative NAFDI index values. Phase II from 24 June to 24 August, which is dominated by positive NAFDI index values. And Phase II from 24 to 31 August characterised by negative index values again. To demonstrate the pressure distribution over North Africa corresponding to negative respectively positive NAFDI index values, composite plots

for the 25th percentile and 75th percentile are compiled. Figures 4b and 4c present the composites for the 850hPa geopotential calculated on the basis of these days, on which the NAFDI index is within the range of the 75th percentile and 25th percentile respectively. The 850hPa level is chosen as winds on this level impact on both, dust emission and dust transport. The 75th percentile represents the particular positive NAFDI values, the 25th percentile particular negative NAFDI values. The JJA 2013 index median is slightly positive, 0.9. NAFDI index values above 2.7 are included in the 75th percentile, index values below

-3.1 are included in the 25th percentile.The composites are based on the quartiles to achieve a clear contrast between the two phases. A similar approach is suggested by *Rodriguez et al.* (2015). Low NAFDI index values are associated with a SHL in its western position, generally low pressure over the Sahara, and a weak subtropical ridge over the Mediterranean basin. In contrast to this, high NAFDI index values reflect a pressure distribution characterised by a SHL in its eastern position, and a pronounced subtropical ridge over the Mediterranean basin with an enhanced pressure gradient towards the SHL.

Similar to the distribution of the geopotential reflecting the atmospheric circulation at low level, composites are calculated for the atmospheric dust loading expressed as dust AOD (Figs. 5a and b) and the meridional wind components (Fig. 6). The AOD composites for low and high NAFDI index values address the question for a link between atmospheric dust load and NAFDI phase. To illustrate the difference in AOD during particular low respectively high NAFDI phases to the JJA average, the relative differences are shown in Figs. 5c and d. As the NAFDI is based on the pressure gradient and so is the wind determining dust

entrainment and transport, composites of the meridional wind speed complement the relation between NAFDI and dust AOD. In order to distinguish between north-easterly Harmattan and south-westerly Monsoon inflow, the meridional wind components are considered separately for northerly (representing Harmattan conditions) and southerly (representing Monsoon conditions) winds. The composite shows the average wind speed for the corresponding wind direction over the selected time period. In agreement with a weaker Harmattan for periods characterised by low NAFDI index values (Fig. 6), the dust plume over the

Sahara extends further towards and into the Mediterranean basin as southerly winds prevail (Fig. 5). As the pressure gradients are weaker, less dust is suspended in the atmosphere resulting in lower dust AOD values. During periods characterised by positive NAFDI index values, the atmospheric dust loading over the central Sahara at about 20-25°N is increased and so is the dust AOD over the dust outflow region off the West African coast. This situation is complemented by the dust export flux (Fig. 7), which shows that the core of the dust flux at 20°W is shifted to the south for days characterised by a particular

low (negative) NAFDI index. Furthermore, the re-circulation of dust resulting into positive (eastward) dust fluxes at 20°W is enhanced compared to the JJA 2013 mean. The dust export (westward dust flux) is enhanced for days characterised by a positive NAFDI index representing strong Harmattan winds.



## 6.2 Saharan Heat Low

The low-level atmospheric thickness (LLAT) between 925hPa and 700hPa is used to identify the SHL as suggested by *Lavaysse et al.* (2009). Thereby, the highest 10% of the LLAT values over North Africa (0-40°N, 20°W-30°E) are defined as the SHL. During June and July, the centre of the SHL migrates from a mean position south-west of the Hoggar Massif

to a more north-westerly position between the Atlas Mountains and the Hoggar Massif, where it remains quasi-stationary throughout July and August. The depth of the SHL increases throughout June and July reaching its maximum in July. The spatio-temporal evolution of the depth, extent and geographical position of the centre can be summarized by a Hovmöller plot as shown in Fig. 8a. The maximum depth of the LLAT indicating the SHL is illustrated as time series in Fig. 8b. The individual LLAT maximum values show little variability over the course of the summer (JJA), as the SHL itself exists over the entire

period over North Africa. However, the geographic position of the centre of the SHL is migrating. The SHL is characterized by a more southerly position during June and deepens throughout June reaching its maximum in 2013 during the second half of July (23 July 2013, LLAT = 2720m) as expected. The deepening and north-westward propagation of the SHL is of intermittent nature showing phases of increasing depth and northward propagation and phases of a southward retreat and decreasing intensity. Throughout the June to August 2013 period, the LLAT indicating the depth and thus the strength of the SHL varies

around an average value of 2699m with a standard deviation of 77m. During JJA 2013, two phases of a particular deep SHL occurred: A first but weaker period of LLAT maxima above the 75-percentile (2706m) occurred during 18 to 23 June, although the LLAT maxima did not reach the similar high values, mainly as this period is earlier during the summer season. A second and significantly stronger phase occurred during 14 to 28 July 2013, the period which also includes the overall maximum LLAT depth for that summer. The LLAT maximum values were continuously above the 75th percentile, expect for one day (21 July).

The second, stronger SHL phase is framed by two periods of a significantly shallow SHL with LLAT maximum values below the 25th percentile (2692m) during 9 to 11 July and 31 July to 13 August. The latter phase was briefly intermitted by three days of increased LLAT values, however, the generally shallow extend of the SHL dominated.

The dust AOD distribution varies among the different strengths of the SHL, which is illustrated in Fig. 9 for the 75th percentile (a) and 25th percentile (b) of the LLAT respectively. A particular deep and strong SHL (75th percentile) is accompanied by

higher dust AODs over the central Sahara compared to a particular shallow and therefore weak SHL, which is accompanied by a generally lower AOD levels. Regarding the dust export towards the east Atlantic, the dust outflow region is slightly shifted to the south for the stronger SHL condition, however, the dust AODs corresponding to the dust outflow region are at a comparable level. Comparing the dust AOD distribution for these two contrasting SHL conditions, significant differences are evident. The differences are strongest north of 35°N, where the dust AOD level is significantly enhanced over the Atlas region due to the

unusual northward extend of the Saharan dust plume during strong SHL conditions (Fig. 9a and c). Also, the level of dust AOD is enhanced over the southern Sahara and Sahel zone (south of 20°N). This may be due to stronger winds along the southern margin of the SHL caused by an enhanced pressure gradient. During conditions with a particular weak SHL (25th percentile), the level of dust AOD over North Africa is generally lower, except for an intermittent region at around 20°N. Particularly low





AOD levels are evident over coastal Libya, associated with the generally low atmospheric dust load over that region (Fig. 9b and d). This may be due to the more southerly position of the SHL during these periods.

### 6.3 West African Monsoon

The West African Monsoon circulation is driven by the Hadley-circulation and supported by the gradients in temperature and humidity resulting from the contrast between the hot and dry Sahara and the humid and cooler Gulf of Guinea (*Parker et al.*, 2005a). To illustrate the intermittent nature of the northward propagation of the monsoonal air, Fig. 10 shows the northward propagation of the humid monsoonal air summarized by the Hovmöller diagramm. The northward transport of monsoonal, humid air through 20°N is expressed as the northward flux of humidity, calculated as product of the specific humidity and the northward wind speed at near-surface level (10m above ground). At 20°N, the monsoonal air arrives in pulses, which reflect the intermittent character of the northward pushes of the monsoon front and illustrates the non-stationarity of the WAM circulation. The observed intermittency agrees with the impact of the passage of AEW and associated troughs over this region. However, a clear relation between the intensity or frequency of the monsoon front's northward propagation and the NAFDI or LLAT index is not apparent. The temporal evolution rather illustrates the pre-monsoon phase until end of June and the onset of the monsoon in July. During its peak, monsoonal air propagates as far north as 20°N. Figs. 6c and d illustrate the monsoonal (southerly) winds over North Africa on days corresponding to positive NAFDI respectively negative NAFDI index values. Although including no information on the actual humidity of the air mass, the meridional wind direction and thus the origin of the air mass implies the predominance of monsoonal air. E.g., on days characterised by particularly negative NAFDI index values, the southerly winds are apparent south-west of the central Saharan mountains. This region is also identified as major dust source region during the second half of the day (cf. Sec. 4 and Fig. 2) probably activated by deep-moist convection embedded in the monsoonal air.

### 7 Discussion and implications for dust transport and export

Dust transport and dust export are closely related. Both are describing the flux of dust within the air: Prevailing wind regimes determine transport pathways, and deposition respectively sedimentation velocities determine dust removal rates from the atmosphere. In the context of this study, dust export is considered as the dust transport towards the Ocean and thus off the North African continent, e.g. through 20°W as a simplified representation of the West African coast line. The zonal flux $M_z$ is calculated as the product of zonal wind velocity $v$ and dust concentration $c$:

$$M_z = v \cdot c \qquad (2)$$

The objective of this study was set to investigate the atmospheric controls on the atmospheric dust life-cycle with particular focus on the dust export towards the Atlantic and ultimately towards the Caribbean Sea. Apparently, dust emission and transport pathways over the North African continent determine the dust export. The amount of dust crossing the 20°W meridian





is characterised by a function of latitude, vertical height and time describing the intermittent nature of the dust flux towards the eastern north Atlantic. Figure 11 presents its variability over time for June to August 2013. The figure panels (Fig. 11a and Fig. 11b) are complementary: Figure 11a illustrates the predominant height of dust transport, but does not distinguish for different latitudes. Figure 11b pictures the variability of the vertically integrated dust transport fluxes over the different latitudes

disregarding the variability in transport height.

The height of dust transport through 20°W is crucial for the cross-Atlantic transport. Dust transported within the Saharan Air Layer (SAL), which is situated above the trade wind inversion at heights between 800 and 500hPa is often described as the transport-highway for dust across the Atlantic. Whereas dust transported within the marine boundary layer shows enhanced dust removal rates, and consequently high deposition fluxes, dust transport within the SAL is more efficient and higher dust

concentrations are more likely to reach the Caribbean. Generally, dust export within the SAL is characteristic for the summer season (e.g., *Schepanski et al.*, 2009a).

Bringing together Figs. 11a and 11b, JJA 2013 dust transport through 20°W occurred predominantly within the SAL and showed an intermittent character, which is typical for summertime dust export (*Jones et al.*, 2003; *Schepanski et al.*, 2009a). The intermittent character of dust plumes or dust pulses leaving the North African continent and entering the Atlantic becomes

also evident when considering the dust flux through 20°W as a function of latitude and time (Fig. 11b).The major fraction of dust mass being transported westwards is leaving North Africa between 12 and 23°N. Re-circulating dust plumes may enter the North African continent again to the north or south of the SAL core, particularly at around 25°N.

This study aimed at investigating the North African dust outflow and consequent dust export towards the north Atlantic depending on the atmospheric circulation present over North Africa. To cluster similar circulation regimes characterised by different

strengths of the major atmospheric elements, namely Harmattan, SHL, and WAM, different indices were presented (cf. Sec. 6). Figure 12 presents the temporal evolution of the Harmattan circulation represented by the NAFI index and the LLAT representing the SHL together with the westward (negative values) an eastward (positive values) dust flux through 20°W. We focus on the NAFI and LLAT only as these two indices are directly linked to the wind regime over the North African continent. The humidity flux used as indicator for the northward propagation of the WAM is not included here as it is not directly representing

the pressure respectively wind field resulting into dust emission and dust transport. As the individual time series for NAFDI, LLAT, and dust flux are presented above, the coincidence of individual phases of particular high or low index values are in focus. Generally, the Pearson correlation *r* between NAFDI respectively LLAT and eastward or westward dust flux is close to zero (r(NAFDI, eastward dust flux) = -0.04, r(NAFDI, westward dust flux) = -0.3, r(LLAT, eastward dust flux) = 0.3, r(LLAT, westward dust flux) = -0.1) and thus not significant. Overall and despite the intermittent character of the dust export flux, the

westward dust flux is enhanced during the period characterised by positive NAFDI index values, which represent a distinct Harmattan due to an enhanced pressure gradient across West Africa. Comparing dust flux and SHL intensity represented by the LLAT, it is noticeable that the pronounced reduction in westward dust flux coincides with the sudden reduction of the LLAT on 21 June 2013. Enhanced levels of westward dust export fluxes match with a diminishing LLAT. The maximum westward dust export matches with the diminishing phase following the LLAT maximum. Nevertheless, the correlation is below level





of confidence and suggests that the relation between NAFDI, LLAT and dust flux is not as robust as expected. One reasons explaining this may be the temporal lag at which the Harmattan winds and SHL intensity act on the dust export.

# 8 Conclusions

Harmattan, Saharan heat low, and West African Monsoon are the three major elements of the atmospheric circulation over North Africa, the most active and most contributing dust source region on Earth. The source regions are active throughout the year and a constant outflow of dust leaving the continent can be observed. During summer, a continuous flow of dust crossing the North Atlantic towards the Caribbean is apparent and described by the Saharan Air Layer. However, quasi on top of this background flow of dust, intermittent events of enhanced dustiness occur. They are described by dust plumes or pulse of dust that can be impressively visualized by satellite images. This study aimed at investigating the impact of the three major elements determining the atmospheric circulation over North Africa on the variability of the dust outflow, which can be understood as an amplification over the background flow. To achieve this, the variability of each element was described by different methods described in the literature. The temporal variability of the strength of the Harmattan was described by the NAFDI index (*Rodriguez et al.*, 2015), the Saharan heat low was described by the LLAT (*Lavaysse et al.*, 2009), and the West African Monsoon inflow was identified by corresponding moisture flux as shown by *Parker et al.* (2005a). The dust flux through $20°N$ was defined here as the ultimate measure to quantify the temporal variability of the dust export towards the north Atlantic. All three methods describing the variability of the individual circulation element were compared against the variability in the dust flux. Although each method identified phases of enhanced and weak activity of the corresponding meteorological element, the temporal agreement among them is only evident for individual periods. This may be due to the time-scale of the study, which focussed on daily effects rather than climatological evidence or tendencies. Certainly, findings from the present study are limited to the specific situation of summer 2013. However, the results suggest to consider the impact of the elements of the atmospheric circulation over North Africa on the dust export as a modulation on top of the background dust outflow, in which the effects of the individual elements are supposed.

# 9 Data availability

COSMO-MUSCAT data are available on request. The dust source activation frequency (DSAF) data set is published as doi:10.14759/41916.2016.1. AERONET coarse mode AODs are available from http://aeronet.gsfc.nasa.gov.

*Acknowledgements.* KS acknowledges funding through the Leibniz Association for the Project "Dust at the interface - modelling and remote sensing". The authors thank the Deutscher Wetterdienst (DWD) for cooperation and support, and the AERONET team from the sites at Santa Cruz de Tenerife, Capo Verde, Dakar, Cinzana and Banizoumbou for obtaining the measurements and providing the data.



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



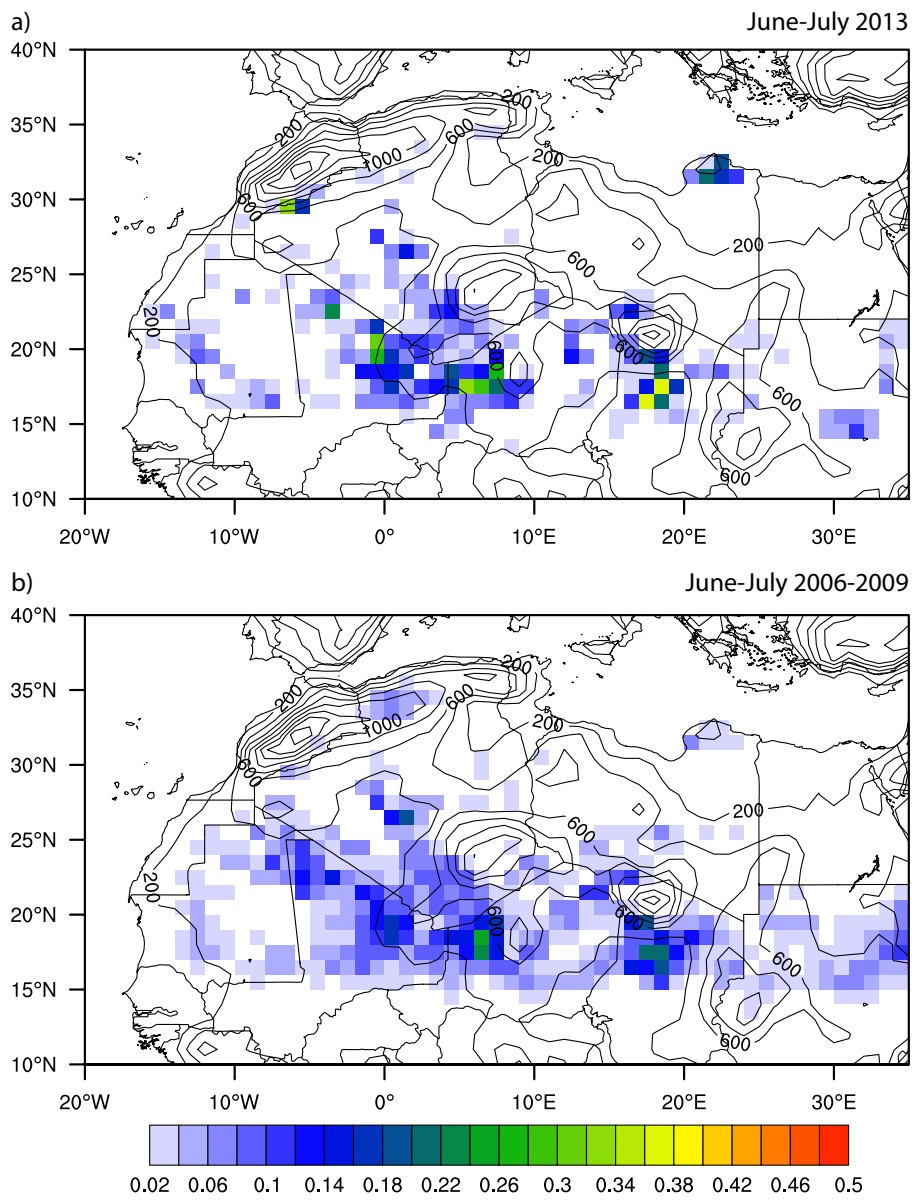

**Figure 1.** Occurrence frequency of dust source activation (fraction of days with dust occurrence) for the two-month period June and July for 2013 (a) and the four-year period 2006-2009 (b). Topography [m] is given by the black solid lines at 200m intervals.




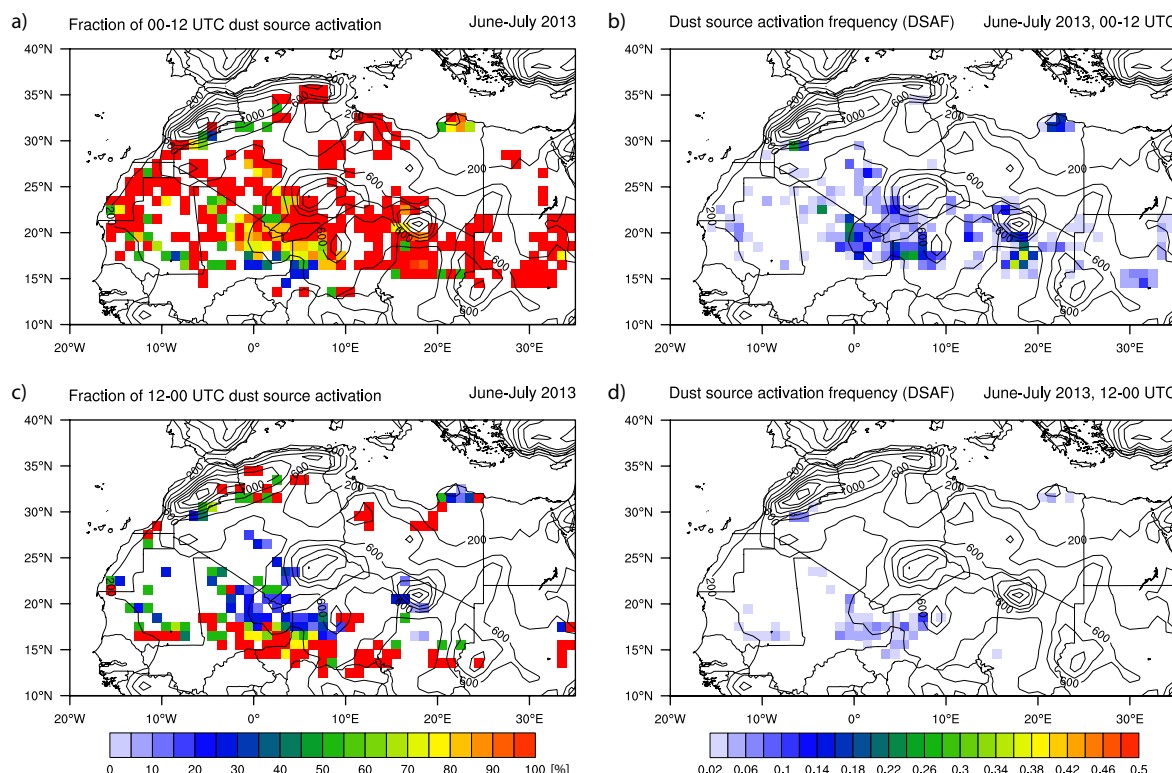

**Figure 2.** Occurrence frequency of dust source activation (fraction of days with dust occurrence) for the two-month period June and July 2013 for the first half of the day (a, b) and the second half of the day (c, d) expressed by fraction of the total (a, c) and occurrence frequency (b, d).



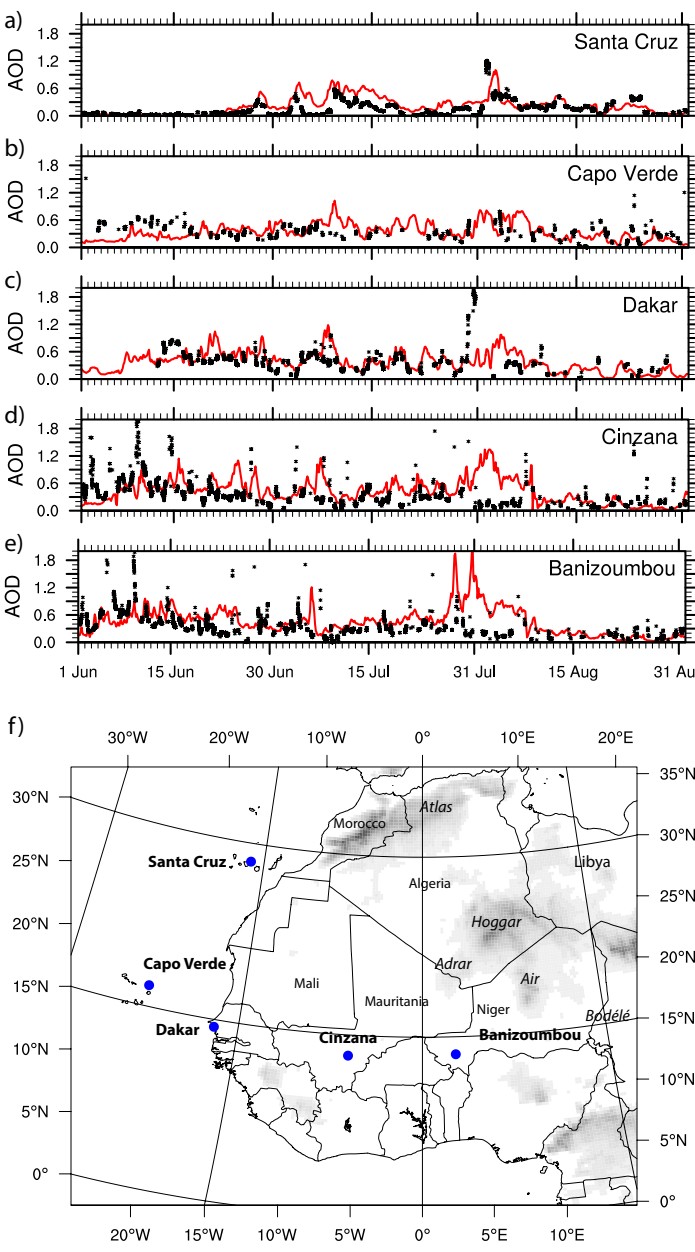

**Figure 3.** Dust AOD calculated from COSMO-MUSCAT dust concentration fields (red) and coarse mode AOD estimated from AERONET sun-photometer measurements (black) for JJA 2013 at five different station: (a) Santa Cruz (28.473°N 16.247°W, Spain), (b) Capo Verde (16.733°N 22.935°W, Cape Verde), (c) Dakar (14.394°N 16.959°W, Senegal), (d) Cinzana (13.278°N 5.934°W, Mauritania), and (d) Banizoumbou (13.541°N 2.665°E, Niger). An overview on the stations' geographic location and geographic names (AERONET site: bold, geographic names: italic, countries: regular) used in this publication are given in (f); orography is implied by grey shading.



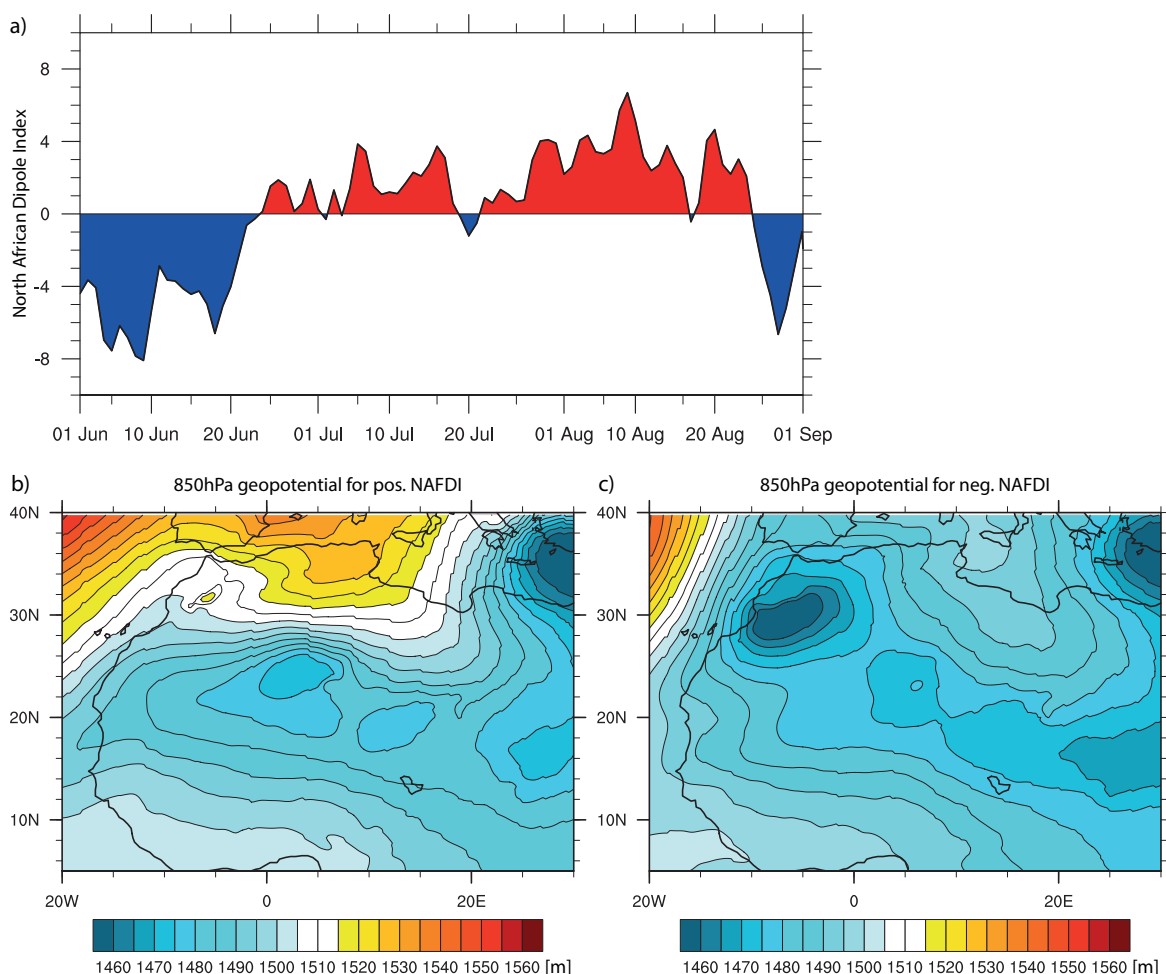

**Figure 4.** (a) North African Dipole Intensity index calculated following Eq. 1 reflecting the strength of the Harmattan flow over North Africa. (b) Composite of 850hPa geopotential height [m] days within the 75th percentile range of the NADI index values, and (c) for days within the 25th percentile range.







**Figure 5.** Composite of dust AOD for days within the 75th percentile (a) and 25th percentile of the NAFDI index values. (c) and (d) show the relative difference to the JJA 2013 mean.







**Figure 6.** Composite of meridional winds for 75th percentile (positive values) and 25th percentile (negative values) of the NAFDI index values. The meridional winds [m/s] are shown separately for northerly winds referring to the Harmattan (a, b), and southerly winds referring to the Monsoon (c, d).



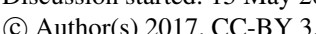



**Figure 7.** Composite plots of the vertically integrated zonal dust flux [kg /m$^2$] at 20°W for the days out of June, July, August 2013, which correspond to the 75th percentile (a) and 25th percentile (b) of NAFDI index values. Dust export towards the North Atlantic is represented by negative values, positive values indicate transport back to the North African continent. Dust fluxes are totals over the 10-25°N latitudinal band. Differences in dust flux compared against the JJA 2013 mean dust flux are shown in (c) and (d).



**Figure 8.** (a) Hovmöller diagram of low-level atmosphere thickness (LLAT) [m] representing the SHL at 23°N. (b) Time series of maximum LLAT [m] for June to August 2013. Blue circles indicate daily LLAT maximum values less than the 25th percentile, red circles indicate daily LLAT maximum values above the 75th percentile. The LLAT is calculated from COSMO-MUSCAT simulation following *Lavaysse et al.* (2009).







**Figure 9.** Composite of dust AOD for the days corresponding to the 75th percentile (a) and 25th percentile (b) of the LLAT. See further Fig. 8 for the assignment of the days to the 25th and 75th percentile respectively. (c) and (d) show the relative difference to the JJA 2013 mean.




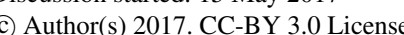

**Figure 10.** Meridional monsoon flux [kgm/kgs] at 20°N at surface level calculated as the product of the northward component of the meridional wind and the specific humidity, both at 10m above ground level.





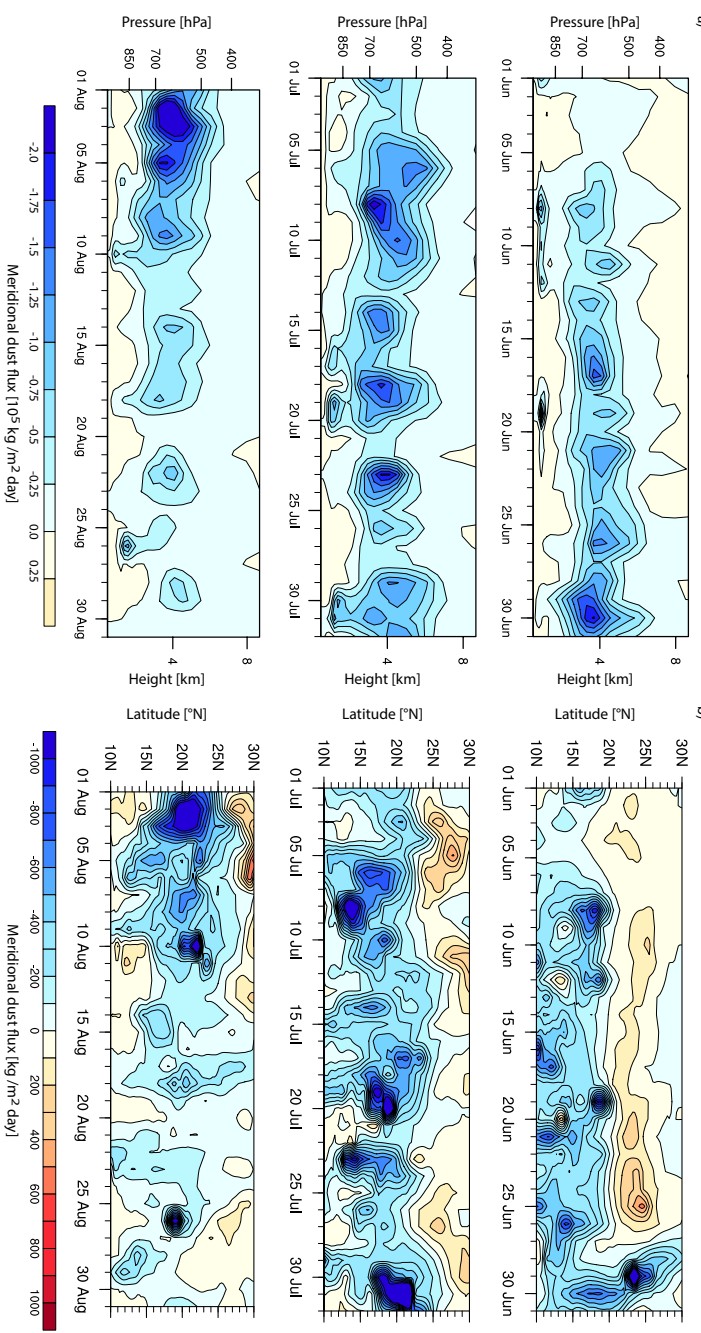

**Figure 11.** (a) Zonal dust flux [kg /m² day] at 20°W. Dust export towards the North Atlantic is represented by negative values, positive values indicate transport back to the North African continent. Dust fluxes are totals over the 10-25°N latitudinal band. (b) Zonal dust flux [kg /m²day] at 20°W. Dust export towards the North Atlantic is represented by negative values, positive values indicate transport back to the North African continent. Dust fluxes are totals over the vertical column (0-22km).




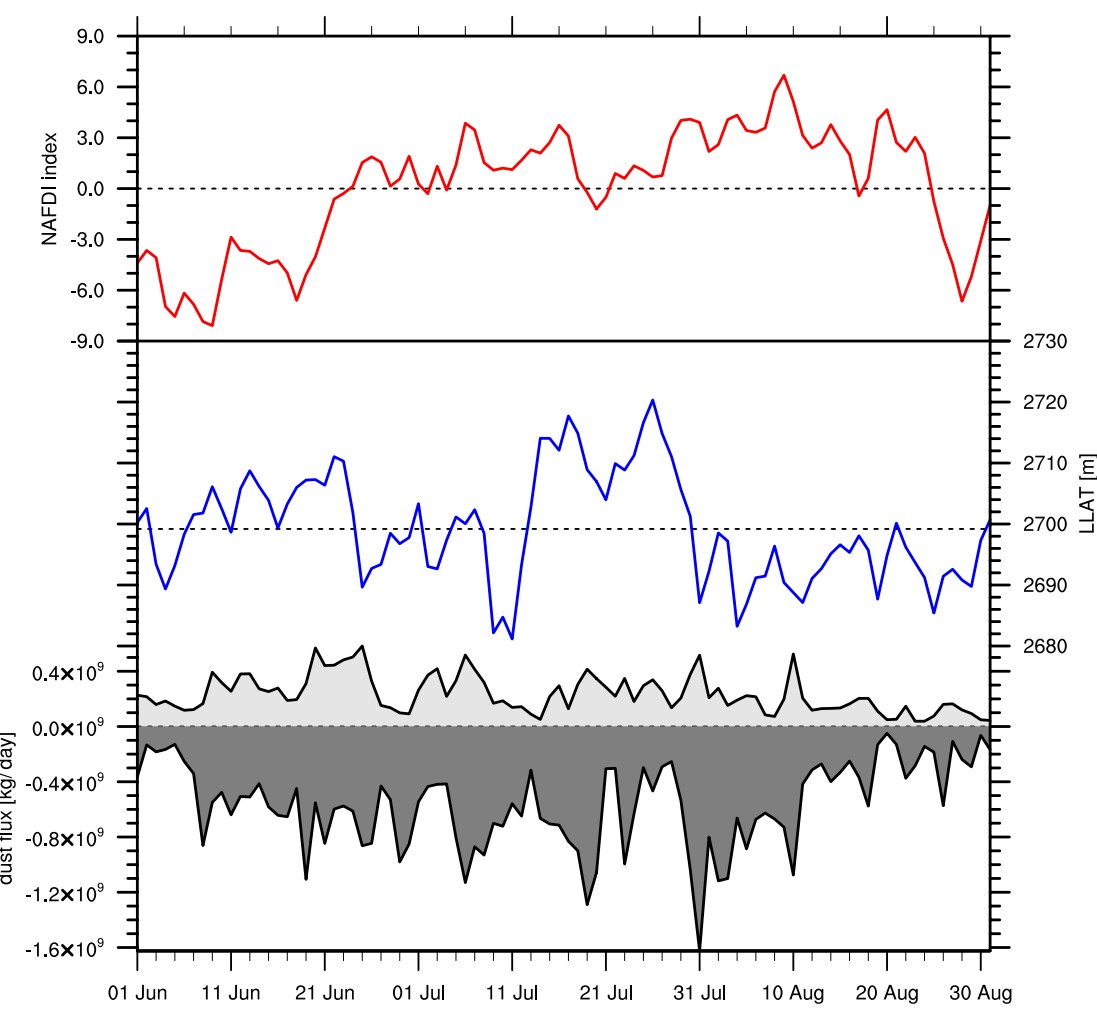

**Figure 12.** Time series of the NAFDI index (red) indicating the strength of the Harmattan, the LLAT (blue) representing the depth of the SHL, and the westward (dark grey) and eastward (light grey) dust flux at 20°W integrated over the total column (0-22km altitude) and the latitudinal band between 10 and 25°N.