# Peer review of "Harmattan, Saharan heat low and West African Monsoon circulation: Modulations on the Saharan dust outflow towards the north Atlantic"

_Atmospheric Chemistry and Physics, 2017_

## Referee Comment (RC1) · Anonymous Referee #1 · 26 Jun 2017

Review of the paper "Harmattan, Saharan heat low and West African Monsoon circulation: Modulations of the Saharan dust outflow towards the north Atlantic" by Schepanski et al.

This study aims at assessing the influence of the main components of the West African Monsoon on the Saharan dust outflow towards the north Atlantic. After a careful reading, I am afraid to request major revisions before to accept this study. Therefore, despite some interesting results to study the mechanism associated with dust outburst, the second part (from section 6) of the study is not clear: the objectives are not well identified, the methodology is not well adapted and some conclusions are too specu-

lative. For these reasons, I would recommend to deeply modify the second part that includes a clarification of the objectives to the improvement of the methods. The major and detailed comments are provided below.

Major comments

Why the climatology is restrained as the period 2006-2009?

The figures 1 and 2 are done using June-July then in the second part JJAS is considered. This creates some confusions.

Once the dust outbursts for the year 2013 are analyzed, the second part (from section 6) is not clear to me. What is the objective? As I understand, the impacts of the seasonal cycle of the main components of the West African Monsoon are studied. But in West Africa, all these components are governed by a strong and similar (or at least very close) seasonal cycle. For example, the NAFTI index is clearly related to the onset, established and retreat monsoon phases.

Why only the NAFTI and the SHL are specifically analyzed? The monsoon flow is mentioned (Fig. 10) but why there is no analyze of its impact as done for the others components?

The SHL is defined following a method proposed by Lavaysse et al. (2009), where the SHL moves depending the LLAT intensities. The intensity of the SHL is thus one part of its characteristics, and the location of the SHL may have strong impacts depending where it is located. What happen if the longitude or the latitude is chosen? To avoid the problem related to the transitions phases of the SHL in June and September, the mean LLAT over the Sahara could be used.

Detailed comments

p5 l2 double brackets

p6 l7 Chauvin et al. (2010)

[Figure]

p6 l19 10% represents a spatial or temporal detection?

p8 l15-l23 There is no dust coming from outside the domain (Arabian Peninsula)?

p10 l7 and Fig. 2 It could be interesting to compare these results with the climatology (as done for Fig. 1)

p11 last paragraph of section 5 and Fig. 3: Scatter plots are more suitable for validation purposes. Bias, correlation and uncertainties are better represented.

p11 l29 The low NAFTI index value is defined according to the climatology?

Figures:

Fig 1: Add Bodele region in the map, please adjust the color scale

Fig 2: adjust the color scale. Also the change in % is not necessarily well adapted since it overestimates the region with low dust activation.

Fig 3: Please change to scatter plots

Fig 5: Again, I am not convinced by the scales used (in %). This increases the weight of the signal over region with low AOD. The difference is more adapted to me.

Fig 6: Very complicated to understand how this figure is realized. Why the Q25 and 75 of U or V are not simply used.

Fig 7: This figure looks like a simple difference in between the established and the onset (or retreat) phases and do not bring innovative information

Fig 9: Same problem with the scale in %

Fig 12: Why the intensity of the monsoon is not added?

---

## Author Comment (AC1) · 2 Jul 2017

*The authors would like to thank the reviewer for the time spend on the manuscript, and for providing helpful and constructive comments and suggestions. We have considered carefully all comments made; please find our detailed reply (italic) below.*

**Anonymous Referee 1**

This study aims at assessing the influence of the main components of the West African Monsoon on the Saharan dust outflow towards the north Atlantic. After a careful reading, I am afraid to request major revisions before to accept this study.

[Figure]

Therefore, despite some interesting results to study the mechanism associated with dust outburst, the second part (from section 6) of the study is not clear: the objectives are not well identified, the methodology is not well adapted and some conclusions are too speculative. For these reasons, I would recommend to deeply modify the second part that includes a clarification of the objectives to the improvement of the methods. The major and detailed comments are provided below.

**Major comments**

Why the climatology is restrained as the period 2006-2009?

*The DSAF for June, July 2013, during which the SALTRACE field campaign took place, are compared to June, July 2006-2009 (4 years) as the DSAF analysis is available for just these years, and cannot be easily expanded. Nevertheless, while a certain year-to-year variability is evident in the DSAF analysis the individual years agree clearly in the locations of major dust source areas as well as dependence of the dust events on meteorological features, such that we find the use of a climatology from 4 years as a baseline to be acceptable for the purpose of this publication.*

The figures 1 and 2 are done using June-July then in the second part JJAS is considered. This creates some confusions.

*The satellite-based DSAF are shown for June, July only as the SALTRACE field campaign took place during these months. Based on the DSAF analysis, the model-based approach was developed. In order to draw a conclusion, which is representative for the entire season, the considered period was expanded to include June, July, and August (JJA). We did not consider September at all in the context of this study.*

Once the dust outbursts for the year 2013 are analyzed, the second part (from section 6) is not clear to me. What is the objective? As I understand, the impacts of the seasonal cycle of the main components of the West African Monsoon are studied. But in West Africa, all these components are governed by a strong and similar (or at

least very close) seasonal cycle. For example, the NAFTI index is clearly related to the onset, established and retreat monsoon phases.

*The focus of the present manuscript is on the variability of the atmospheric circulation over North Africa governed by the different circulation elements and their impact on North African dust export towards the tropical north Atlantic. These circulation elements or features are: (1) the Harmattan fow, (2) the Saharan heat low, and (3) the West African Monsoon. As stated in the introduction, the aim of this study is twofold. "First, it discusses the spatio-temporal distribution of active dust sources over North Africa regarding implicit information on the meteorological elements driving dust emission and their predominance during a typical summer season. To achieve this, the study makes use of satellite observations for identification of active dust source regions. Second, the study elaborates the atmospheric circulation with regard to its relevance for dust source activation and dust transport towards the tropical North Atlantic. For this, simulations from the dust-atmosphere model system COSMO-MUSCAT are analysed."*

*We think the review too much stresses the role of the West African Monsoon. However, as also shown by the outcomes from this study, the West African Mosnoon system is more complex. The strength of the Harmattan flow, the nature of the Saharan heat low, and the propagation of the Monsoon front are linked and in concert determine the atmospheric circulation over North Africa. The North African Dipole Intensity (NAFDI), which simply is a measure for the pressure gradient across North Africa, reflects the strength of the Harmattan flow and the westward displacement of dust-laden air within the Saharan Air Layer. Therefore, the NAFDI is used as an easy-to-understand proxy index for the Saharan dust export towards the North Atlantic Ocean.*

*In contrast, the monsoon onset (retreat) is a less appropriate indicator for describing the modulations on the Saharan dust export due to the lack of a concise definition. See Fitzgerald et al. (J.Clim., 2015): "The concept of West African monsoon onset is not a straightforward issue. While the monsoon system is clearly defined, picking a singular point at which onset occurs is reliant on understanding what onset means to*

*the end users."*

Why only the NAFTI and the SHL are specifically analyses? The monsoon flow is mentioned (Fig. 10) but why there is no analyse of its impact as done for the others components?

*The focus of this manuscript is on the atmospheric circulation modulating the North African dust export towards the tropical North Atlantic. The Harmattan circulation and the SHL are, due to their geographical position, prominent features of the atmospheric circulation stimulating or even determining North African dust emission and consequent dust transport. The direct impact of the strength of the monsoon circulation on the dust export towards the north Atlantic appears to be minor compared to the impact of the Harmattan flow. Nevertheless, the SHL as an element stimulated by both the Harmattan flow and the Monsoon flux, indirectly reflects the impact of the Monsoon circulation on the atmospheric circulation over North Africa ultimately modulating the dust export towards the north Atlantic.*

The SHL is defined following a method proposed by Lavaysse et al. (2009), where the SHL moves depending the LLAT intensities. The intensity of the SHL is thus one part of its characteristics, and the location of the SHL may have strong impacts depending where it is located. What happen if the longitude or the latitude is chosen? To avoid the problem related to the transitions phases of the SHL in June and September, the mean LLAT over the Sahara could be used.

*The role of the longitude on the position of the SHL identified via LLAT is presented in Fig. 8. The spatio-temporal evolution of the SHL was plotted in various ways including Hovemöller diagrams at different longitudes and latitudes. Eventually, we came to the conclusion that the in Fig. 8 shown Hovemöller diagram presenting the longitudinal propagation along $23°$ N is appropriate for illustrating variabilities in SHL strength and position modifying Saharan dust transport. As we focus on the variability in the SHL modulating North African dust transport, we prefer to use daily values for the SHL*

*position, strength, and extend over the multi-monthly mean.*

**Detailed comments**
p5 l2 double brackets
*This comment refers to the following text: "(June, July, August (JJA))". As there are two left parentheses used, we also use two right parentheses. No changes will be made to the revised version of the manuscript.*

P6 l7 Chauvin et al. (2010)
*Many thanks for spotting this. It's corrected in the revised version of the manuscript.*

P6 l19 10% represents a spatial or temporal detection?
*"The area of the SHL is refined to the 90th percentile of the LLAT values." This means that the area of the SHL is where the local LLAT value exceeds the 90th percentile. In other words: the local LLAT value is part of the upper 10% of the range of LLAT values.*

P8 l15-l23 There is no dust coming from outside the domain (Arabian Peninsula)?
*Correct, there is no dust transported into the domain. Dust originating from the Arabian Peninsula is not considered in this study here as no significant amount of dust originating from the Arabian Peninsula is expected to enter the model domain and contribute to the dust export towards the northern Atlantic. This argument is justified by the prevailing wind regimes over both North Africa and the Arabian Peninsula.*

P10 l7 and Fig. 2 It could be interesting to compare these results with the climatology (as done for Fig. 1)
*Many thanks for this suggestion, which is considered for the revised version of the manuscript. The fraction of dust source activations during the first respectively second half of the day for the 4-year period June, July 2006-2009 shows a similar distribution*

*than the June, July 2013 period: morning-time dust source activations dominate over
the Sahara and Soudan zone (sub-Sahara), where the Harmattan regime and linked
LLJ related winds foster dust emission. Dust source activations related to (moist)
convection predominantly occur over regions dominated by the Monsoon circulation.
Due to the interannual variability, in particular for the number of 12-00 UTC events, the
mean distribution is smoother for the 4-year period than the 1-year period (June, July
only).*

p11 last paragraph of section 5 and Fig. 3: Scatter plots are more suitable for
validation purposes. Bias, correlation and uncertainties are better represented.
*We choose to present the comparison between sun-photometer AOD estimates and
model simulated AOD as time series as this allows for expressing not only the match
of the actual values but also for expressing the match in temporal evolution. This way,
the model's ability to capture individual dust events with regard to strength (dustiness)
and temporal evolution can be validated. This plot type much better serves the study's
objective to analyse the temporal modulations of the Saharan dust export.*

P11 l29 The low NAFTI index value is defined according to the climatology?
*Low NAFDI index values are with regard to the range of values during JJA 2013.
This will be clarified in the revised version of the manuscript: "The JJA 2013 summer
season is characterised by for this period low NAFDI index values during the first three
weeks."*

**Figures:**
Fig 1: Add Bodele region in the map, please adjust the color scale
*The Bodélé region is already shown. The colour scale is adapted to the range of values.*

Fig 2: adjust the color scale. Also the change in % is not necessarily well adapted
since it overestimates the region with low dust activation.

*The colour scale is adapted to the range of values. We choose to express the change in DSAF as fraction (%) as this illustrates the relative magnitude and thus the predominance of the change with respect to the local level of occurrence.*

Fig 3: Please change to scatter plots
*We prefer to show the comparison between AERONET sun-photometer AOD and model simulated AODs as time series. This way, the ability of the model simulation to capture the temporal evolution of dust events is shown, which is a criterion for model evaluation here.*

Fig 5: Again, I am not convinced by the scales used (in %). This increases the weight of the signal over region with low AOD. The difference is more adapted to me.
*Please see our reply to your comment made above.*

Fig 6: Very complicated to understand how this figure is realized. Why the Q25 and 75 of U or V are not simply used.
*Obviously, there is a misunderstanding. This figure actually shows the 75th and 25th percentiles of the meridional (V) wind component. Northerly components are associated with the Harmattan, southerly with the monsoon flow. We refer to the figure caption. The figure shows the composite of meridional winds for the 75th (top (positive) NAFDI values) and 25th (lowest (negative) NAFDI values). We decided to select the winds for the NAFDI percentiles in order to present the "extreme" situations over the normal average condition.*

Fig 7: This figure looks like a simple difference in between the established and the onset (or retreat) phases and do not bring innovative information
*We disagree here. Again, the results are too much seen only from the aspect of the monsoon. The figure presents composites of the vertical distribution of the dust plumes during days with low NAFDI index values (25th percentile) and high NAFDI*

*index values (75th percentile). The NAFDI represents the strength of the Harmattan circulation, and thus the composites represent the dust distribution during two different atmospheric circulation regimes (cf. Fig. 4b, c). "Low NAFDI index values are associated with a SHL in its western position, generally low pressure over the Sahara, and a weak subtropical ridge over the Mediterranean basin. In contrast to this, high NAFDI index values reflect a pressure distribution characterised by a SHL in its eastern position, and a pronounced subtropical ridge over the Mediterranean basin with an enhanced pressure gradient towards the SHL."*

Fig 9: Same problem with the scale in %
*Please see our reply to your comment made above.*

Fig 12: Why the intensity of the monsoon is not added?
*In this study, of course, we also analysed the mass flux through $20°\,N$ averaged between $20°\,W$ and $20°\,E$ to evaluate the impact of the monsoon flux. The results, however, show that due to its intermittency, of the monsoon flux, which is orientated in meridional direction, the mass flux through $20°\,N$ averaged between $20°\,W$ and $20°\,E$ does not seem to be an appropriate measure to be related with the dust export towards the tropical north Atlantic, which is orientated in zonal direction. Therefore, we decided to not show the integrated monsoonal flux in Fig. 12. Nonetheless, we emphasise that the monsoon impact is implicitly included in LLAT and NAFDI.*

---

## Referee Comment (RC2) · Anonymous Referee #2 · 11 Jul 2017

General remarks:

The present manuscript investigates the atmospheric circulation pattern over North Africa about its role favouring dust emission and dust export towards the tropical North Atlantic. The focus of the study is in Summer 2013 (June to August) when it took place the field campaign SALTRACE (Saharan Aerosol Long-range TRansport and Aerosol-Cloud interaction Experiment). While the results of the study are interesting to be published, their presentation and discussion are not yet sufficient to be published in Atmospheric Chemistry and Physics in the current form. Therefore, it is worth to be published after addressing major revisions which are explained below along with a few

[Figure]

other details.

Major comments:

One of my main concerns is related to the study period, i.e. in summer 2013. This is justified because of the SALTRACE experimental campaign. However, I couldn't find any model comparison within the exceptional observational database from this experimental campaign which includes among others aerosol vertical profiles neither any link to other publications related to this campaign and the link with your results. Then, why are you limiting your analysis to summer 2013? If you include more years, the results would be more representative. If not, it would be reasonable that in the discussion of the results, you also consider to include a description of the vertical dust structure associated with the North Atlantic dust transport (and its associated dust sources) and the relation with the dust concentrations measured in the Caribbean.

Otherwise, it would be good that you reinforce the performance of the model results because the evaluation sounds qualitative. You don't include any performance skill score with the AERONET database neither comparison with satellites. Meanwhile, your analysis of the dust emission is based on satellites; you only use the model results for the analysis of the dust transport. In this sense, how is the agreement between the dust emission fields between satellite and model results? Can the model reproduce the results (timing and spatial distribution) obtained from MSG? For example, from the model evaluation against AERONET is clear that haboobs are missing in your simulations (see Cinzana and Banizoumbou in late July in Figure 3). Then, in your discussion about the results based on the model simulations. Are haboobs negligible?

Minor comments:

Page 3 Line 19: A reference to CV-Project (from University of Aveiro, http://www.cesam.ua.pt/subsites/files/seasonal_variability_of_pm_over_cv_island._iceh_lisbon2012.pdf) is missing.

Page 3 Line 34: Please, you include further information about the referenced companion paper.

Page 5 Line 19: Numeric labelling in the title of the section is missing.

Page 10 Line 7: How is the agreement between the dust emission fields between satellite and model results?

Page 10 Line 23: What AERONET dataset are you using? Quality-assured?

Page 11 Line 6: In the AERONET comparison, what about the model overestimations observed in Cinzana and Banizoumbou in early August? Could you include spatial verification of the model outputs based on satellites such as MISR or MODIS?

Page 12 Line 22: Add a reference to Figure 6.

Page 16 Line 2: As you indicate, "maybe there is a temporal lag at which the Harmattan winds and the SHL act on the dust export". Have you been tried to correlate NAFDI/LLAT and dust flux introducing delayed days between them?

Figure 4: Correct NADI by NAFDI.

Figure 11: Indicate in the caption the order of the month for each panel.

---

## Author Comment (AC2) · 31 Jul 2017

*The authors would like to thank the reviewer for the time spend on the manuscript, and for providing helpful and constructive comments and suggestions. We have considered carefully all comments made; please find our detailed reply (italic) below.*

**Anonymous Referee 2**
The present manuscript investigates the atmospheric circulation pattern over North Africa about its role favouring dust emission and dust export towards the tropical North Atlantic. The focus of the study is in Summer 2013 (June to August) when it took

place the field campaign SALTRACE (Saharan Aerosol Long-range TRansport and Aerosol-Cloud interaction Experiment). While the results of the study are interesting to be published, their presentation and discussion are not yet sufficient to be published in Atmospheric Chemistry and Physics in the current form. Therefore, it is worth to be published after addressing major revisions which are explained below along with a few other details.

**Major comments:**
One of my main concerns is related to the study period, i.e. in summer 2013. This is justified because of the SALTRACE experimental campaign. However, I couldn't find any model comparison within the exceptional observational database from this experimental campaign which includes among others aerosol vertical profiles neither any link to other publications related to this campaign and the link with your results. Then, why are you limiting your analysis to summer 2013? If you include more years, the results would be more representative. If not, it would be reasonable that in the discussion of the results, you also consider to include a description of the vertical dust structure associated with the North Atlantic dust transport (and its associated dust sources) and the relation with the dust concentrations measured in the Caribbean.

*The study is motivated by and carried out in the framework of the SALTRACE project. During SALTRACE, measurements were obtained over the Caribbean. Following a holistic approach accounting for the full atmospheric life-cycle of mineral dust, dust sources and emission as well as the transport pathways are required to be considered in order to assess dustiness and dust involving atmospheric processes over the Caribbean. To provide a concise manuscript despite the broad extent of the topic and the aims of SALTRACE, we decided to present the study by two manuscripts: One examining the origin of dust (in particular dust sources and emission processes) and transport pathways towards the north Atlantic (this manuscript), and one presenting the dust transport across the Atlantic and deposition over the Caribbean (the companion paper by Heinold et al. (in preparation). Although being companion papers, both*

*manuscripts are self-contained.*

Otherwise, it would be good that you reinforce the performance of the model results because the evaluation sounds qualitative. You don't include any performance skill score with the AERONET database neither comparison with satellites. Meanwhile, your analysis of the dust emission is based on satellites; you only use the model results for the analysis of the dust transport. In this sense, how is the agreement between the dust emission fields between satellite and model results? Can the model reproduce the results (timing and spatial distribution) obtained from MSG? For example, from the model evaluation against AERONET is clear that haboobs are missing in your simulations (see Cinzana and Banizoumbou in late July in Figure 3). Then, in your discussion about the results based on the model simulations. Are haboobs negligible?
*Many thanks for your suggestion on validating COSMO-MUSCAT dust source regions against these obtained from MSG. We will include this in the revised version of the manuscript. Please be referred to Schepanski et al. (2016) for a more thorough comparison of the model fields against satellite observations. Haboobs are embedded in the West African Monsoon circulation, which provides the moisture reservoir necessary to form deep convection. Thus, the propagation of the monsoon front relates to the formation of deep convection, an essential precondition for Haboobs. The scope of this manuscript is on the general role of the atmospheric circulation regimes on dust source activation and transport pathways. Local-scale processes are of minor importance here compared to the overall strength of the regime.*

**Minor comments:**
Page 3 Line 19: A reference to CV-Project (from University of Aveiro, link) is missing.
*Many thanks for drawing our attention on this activity at Cape Verde. Unfortunately, we could not find any peer-reviewed reference of this activity that we can refer to.*

Page 3 Line 34: Please, you include further information about the referenced companion paper.

*We have included the reference for the companion paper: Heinold, B., K. Schepanski, D. Gieseler, and I. Tegen, Mixing and Deposition Processes during Transatlantic Transport of Saharan Dust, in preparation for Atmos. Chem. Phys.*

Page 5 Line 19: Numeric labelling in the title of the section is missing.

*Many thanks for spotting this. It will be corrected in the revised version of the manuscript.*

Page 10 Line 7: How is the agreement between the dust emission fields between satellite and model results?

*Many thanks for this suggestion. A discussion on this will be added to the revised version of the manuscript. The agreement between dust source activation frequency (DSAF) calculated from COSMO-MUSCAT and these inferred from MSG SEVIRI generally show similar spatial patterns. Both distributions identify a hot spot in terms of DSAF for the source region south of the Hoggar Massif and in between the Adra and Air Mountains. Also, frequent dust emission over southern Mauritania are evident in both data sets.*

Page 10 Line 23: What AERONET dataset are you using? Quality-assured?

*We are using AERONET coarse mode level 2.0 respectively level 1.5 where level 2.0 is not available. Level 2.0 data are available for Santa Cruz and Dakar, level 1.5 data are available for Cape Verde, Cinzana, and Banizoumbou. This information is added to the revised version of the manuscript.*

Page 11 Line 6: In the AERONET comparison, what about the model overestimations

observed in Cinzana and Banizoumbou in early August? Could you include spatial verification of the model outputs based on satellites such as MISR or MODIS?

*The focus of the manuscript is on the general impact of atmospheric circulation regimes on dust export towards the tropical north Atlantic. Thus, daily case studies are beyond the scope of the manuscript. Please be referred to Schepanski et al. (2016) for a more thorough comparison against satellite data.*

Page 12 Line 22: Add a reference to Figure 6.
*A reference to Figure 6 is added (Fig. 7 in the revised version of the manuscript).*

Page 16 Line 2: As you indicate, "maybe there is a temporal lag at which the Harmattan winds and the SHL act on the dust export". Have you been tried to correlate NAFDI/LLAT and dust flux introducing delayed days between them?

*Indeed, we have calculated the correlation between NAFDI respectively LLAT and the dust export flux, including lags of a varying number of days. However, the correlations are not significant (r=[-0.2,0.2] for lag correlations). This is also discussed in Section 7: "Generally, the Pearson correlation r between NAFDI respectively LLAT and eastward or westward dust flux is close to zero (r(NADI, eastward dust flux) = -0.04, r(NAFDI westward dust flux) = -0.3, r(LLAT, eastward dust flux) = 0.3, r(LLAT, westward dust flux) = -0.1) and thus not significant".*

Figure 4: Correct NADI by NAFDI.
*Many thanks for spotting this. Corrected.*

Figure 11: Indicate in the caption the order of the month for each panel.
*The time axis is chronological and is clearly labeled by day and month. We think that no additional description in the caption is necessary.*